# New Tetrahydroacridine Hybrids with Dichlorobenzoic Acid Moiety Demonstrating Multifunctional Potential for the Treatment of Alzheimer’s Disease

**DOI:** 10.3390/ijms21113765

**Published:** 2020-05-26

**Authors:** Kamila Czarnecka, Małgorzata Girek, Przemysław Wójtowicz, Paweł Kręcisz, Robert Skibiński, Jakub Jończyk, Kamil Łątka, Marek Bajda, Anna Walczak, Grzegorz Galita, Jacek Kabziński, Ireneusz Majsterek, Piotr Szymczyk, Paweł Szymański

**Affiliations:** 1Department of Pharmaceutical Chemistry, Drug Analyses and Radiopharmacy, Faculty of Pharmacy, Medical University of Lodz, Muszyńskiego 1, 90-151 Lodz, Poland; malgorzata.girek@stud.umed.lodz.pl (M.G.); przemyslaw.wojtowicz@stud.umed.lodz.pl (P.W.); pawel.krecisz@stud.umed.lodz.pl (P.K.); 2Department of Medicinal Chemistry, Faculty of Pharmacy, Medical University of Lublin, Jaczewskiego 4, 20-090 Lublin, Poland; robertskibinski@umlub.pl; 3Department of Physicochemical Drug Analysis, Chair of Pharmaceutical Chemistry, Faculty of Pharmacy, Jagiellonian University Medical College, Medyczna 9, 30-688 Krakow, Poland; jakub.jonczyk@doctoral.uj.edu.pl (J.J.); kamil1.latka@uj.edu.pl (K.Ł.); marek.bajda@uj.edu.pl (M.B.); 4Department of Clinical Chemistry and Biochemistry, Medical University of Lodz, Narutowicza 60, 90-647 Lodz, Poland; anna.walczak@umed.lodz.pl (A.W.); grzegorz.galita@umed.lodz.pl (G.G.); jacek.kabzinski@umed.lodz.pl (J.K.); ireneusz.majsterek@umed.lodz.pl (I.M.); 5Department of Pharmaceutical Biotechnology, Faculty of Pharmacy, Medical University of Lodz, Muszyńskiego 1, 90-151 Lodz, Poland; piotr.szymczyk@umed.lodz.pl

**Keywords:** acetylcholinesterase inhibitors, Alzheimer’s disease, molecular modeling, Ellman’s method

## Abstract

A series of new tetrahydroacridine and 3,5-dichlorobenzoic acid hybrids with different spacers were designed, synthesized, and evaluated for their ability to inhibit both cholinesterase enzymes. Compounds **3a**, **3b**, **3f**, and **3g** exhibited selective butyrylcholinesterase (*Eq*BuChE) inhibition with IC_50_ values ranging from 24 to 607 nM. Among them, compound 3b was the most active (IC_50_ = 24 nM). Additionally, 3c (IC_50_ for *Ee*AChE = 25 nM and IC_50_ for *Eq*BuChE = 123 nM) displayed dual cholinesterase inhibitory activity and was the most active compound against acetylcholinesterase (AChE). Active compound 3c was also tested for the ability to inhibit Aβ aggregation. Theoretical physicochemical properties of the compounds were calculated using ACD Labs Percepta and Chemaxon. A Lineweaver–Burk plot and docking study showed that 3c targeted both the catalytic active site (CAS) and the peripheral anionic site (PAS) of AChE. Moreover, 3c appears to possess neuroprotective activity and could be considered a free-radical scavenger. In addition, 3c did not cause DNA damage and was found to be less toxic than tacrine after oral administration; it also demonstrated little inhibitory activity towards hyaluronidase (HYAL), which may indicate that it possesses anti-inflammatory properties. The screening for new in vivo interactions between 3c and known receptors was realized by yeast three-hybrid technology (Y3H).

## 1. Introduction

Alzheimer’s disease (AD) is a neurodegenerative disease that represents one of the great healthcare challenges of the 21st century. Unfortunately, little is known about its cause and no curative treatments are available that can stop or reverse its progression. While AD develops over a long preclinical period of several decades, increasing attention is currently being paid to early-onset AD [1,2,3].

Many lifestyle related factors are believed to have a role in dementia, including cardiovascular risk, diabetes, obesity, physical and mental inactivity, depression, smoking, low educational attainment, and diet; however, most have not been fully explored [4].

The apolipoprotein A4 (APOE4) gene is a major risk factor for AD and has several effects on the disease. Lifetime risk for AD is more than 50% for carriers of APOE4 homozygotes and 20–30% for apolipoprotein A3(APOE3) and APOE4 heterozygotes [5].

The accumulation of abnormally folded Aβ and tau proteins in amyloid plaques and neuronal tangles has been found to be related to neurodegenerative processes in the brains of AD patients, as indicated by the amyloid hypothesis [6].

β-Secretase (BACE1) cleaves amyloid precursor protein (APP), which is followed by the cleavage of γ–secretase, leading to the production of beta amyloid (Aβ) peptide [7,8].

Unfortunately, despite the development of many new drugs and clinical trials, no Aβ-targeting drug has been officially approved by the United States Food and Drug Administration (FDA) for the clinical treatment of AD [9].

Only four drugs are used for the treatment of AD. Three of these, the acetylcholinesterase inhibitors donepezil, galantamine, and rivastigmine, might provide some benefit for patients during the first year of treatment, while the glutamate antagonist memantine is typically used in moderate-to-severe dementia, sometimes in combination with a cholinesterase inhibitor. Hence, new treatments are urgently needed to prevent, delay, or treat the symptoms of AD [10]. Tacrine, the first drug approved by the FDA for the treatment of AD, has shown severe hepatotoxicity and many other adverse effects and was withdrawn from the market [11].

There are two types of cholinesterases to hydrolyze ACh in the central nervous system (CNS): acetylcholinesterase (AChE) and butyrylcholinesterase (BuChE). AChE has 10-fold higher hydrolytic Ach activity than BuChE, and selective inhibition of AChE is effective in AD therapy [12]. There is some evidence to suggest that the use of nonselective cholinesterase inhibitors (inhibition of both AChE and BuChE may be more beneficial to patients with AD than the use of selective AChE inhibitors [13].

The design of multi-target-directed ligands (MTDLs) has recently attracted the attention of AD researchers due to the pathological complexity of AD. The MTDLs strategy in the search for potential anti-AD drugs is very promising because the traditional approach of modulating one target is not very effective. The MTDLs strategy has become very popular over the past several years. It is a combination of a known AChE inhibitor with another moiety to create multifunctional hybrid with beneficial properties in the treatment of AD [14,15,16].

The screening for new in vivo interactions between small molecule ligands and known receptors may be realized by yeast three-hybrid technology (Y3H) [17]. The Y3H is an advancement of the yeast two-hybrid (Y2H) method, where the expression of reporter genes is mediated by the protein–protein interaction between bait and prey proteins [18,19]. In the Y3H method, the hybrid small molecule ligand is built from two parts, known as the hook and fish, which are required to mediate the hybrid protein interactions. The Y3H approach was efficiently used to identify new small molecule ligand interactions with known receptors [20,21,22]. In the presented study, the Y3H approach was applied to test the hybrid ligand-mediated interactions between human acetylcholinesterase (AChE) and four proteins: human amyloid beta A4 protein (A4), human beta-secretase 1A (BACE-1A), human monoamine oxidase B (MAO B), and human microtubule associated protein tau (MAPT).

Inspired by the MTDLs strategy, in our previous works we designed and synthesized a series of novel tetrahydroacridine derivatives as multifunctional agents for the potential treatment of AD [23,24]. These tend to be conjugates of tacrine and 3,5-dichlorobenzoic acid derivatives. To establish whether this chemical modification can impact AD treatment, we synthetized a new series of derivatives. The tacrine-based hybrids are interesting, and new compounds can have complementary pharmacological properties without the toxic effects of tacrine. A novel series may have metal-chelating properties, and in the future, it might be the basis for the design of chemical compounds with potential applications in the diagnosis of AD. The synthesis and biological evaluation of a new series of tetrahydroacridine derivatives that are cholinesterase inhibitors are discussed in this paper. In the present study, the substances were evaluated using Ellman’s assay and beta amyloid aggregation assay. Then, neuroprotective activity was investigated in three experiments: DNA damage, anti-inflammatory properties, and in vivo acute oral toxicity. The mechanism of cholinesterase inhibition was studied by molecular modeling, and the pharmacokinetics of the obtained compounds were predicted using ACD/Percepta.

## 2. Results and Discussion

### 2.1. Chemistry

The new tetrahydroacridine derivatives were synthesized as per Scheme 1. In Scheme 1, firstly **1a**–**1h** compounds were obtained using the reported procedure [25], in which 9-chlorotetrahydroacridine was condensed with diamines in the presence of phenol and NaI at 180 °C for two hours.

Initially, diamine derivatives of tetrahydroacridine (**1a**–**1h**) [26] were bound to 3,5-dichlorobenzoic acid in the presence of tetrahydrofuran (THF), 2-chloro-4,6-dimethoxy-1,3,5-triazine (CDMT), and *N*-methylomorpholine as shown in Scheme 1.

The chemical structures of the newly-synthesized compounds were verified by ^1^H NMR, ESI-MS, HRMS, and IR and are consistent with the proposed structures. Chemical shifts of protons are explained in detail in the Experimental Section.

### 2.2. Biological Evaluation

#### 2.2.1. In Vitro Inhibition Studies on AChE from Electrophorus Electricus (Electric Eel-(*Ee*)) and BuChE from Equine Serum (*Eq*)

The compounds displayed varying inhibitory activity in the percentage inhibition tests. The obtained IC_50_ values for the whole series and reference compound are summarized in Table 1. In particular, most of the new derivatives (**3a**–**3c** and **3f**–**3h**) were found to possess AChE and BuChE inhibitory activity. On the other hand, compounds **3d** and **3e**, with five and six carbons with the alkyl chain, showed lower cholinesterase inhibition (IC_50_ more than 12 and 20 µM inhibition). In addition, most of the compounds were more active as BuChE inhibitors (**3a**, **3b**, **3f** and **3g**) than as AChE inhibitors (**3c** and **3h**). Among the AChE inhibitors, the best was **3c**, with IC_50_ = 25 nM. Additionally, **3c** exhibited lower potential to inhibit BuChE, with IC_50_ = 123 nM. Compounds, **3a** and **3b**, with the shortest carbon chains, were most active inhibitors for BuChE (respectively, IC_50_ = 140 and 24 nM). Compound **3c** was chosen to perform other analyses. Compared to the reference compound in the study, which was tacrine, **3c** showed higher AChE inhibition and lower BuChE inhibition. Therefore, **3c** was more selective for AChE and tacrine for BuChE.

#### 2.2.2. Kinetic Characterization of EeAChE Inhibition

A kinetic study was carried out to determine the inhibition types of the most active compound for AChE (**3c**). A Lineweaver–Burk plot was prepared as shown in Figure 1. All experiments were performed in triplicate. Accordingly, **3c** compound was shown to have mixed-type inhibition.

#### 2.2.3. Inhibition of Self-Induced Aβ_42_ Aggregation by a 3c Derivative

To determine the inhibition of Aβ_42_ aggregation by the new compound (**3c**), thioflavin-T (ThT) assay was performed. Five different concentrations of **3c** were used. At all concentrations, **3c** showed more than 23% inhibitory activity (23% at 5 µM to 32% at 100 µM). The results are summarized in Figure 2. The percentage of inhibition increases slightly as the concentration increases.

#### 2.2.4. In Vitro Evaluation of Potential Hepatotoxicity

Determination of potential hepatotoxicity activity of **3c** was determined by the MTT (3-(4,5-Dimethylthiazol-2-yl)-2,5-diphenyltetrazolium bromidefor) cytotoxicity assay, and tacrine was used as a reference compound. Tacrine possesses hepatotoxicity activity due to the formation of reactive metabolites (for example 7-hydroxytacrine), which appear during tacrine’s metabolism. At concentrations of 10 µM, 1 µM, and 0.1 µM, **3c** demonstrated higher hepatotoxicity (87.37% ± 0.92%, 95.40% ± 0.22%, and 99.87% ± 3.74%, respectively) than tacrine (90.35% ± 3.75%, 99.48% ± 0.74%, and 100.67% ± 1.03%, respectively). One-way ANOVA analysis showed, that there was a significant difference (*p* < 0.05) between results of 1 µM **3c** and 1 µM tacrine. Results of 10 µM and 0.1 µM between **3c** and tacrine did not have significant differences with *p* < 0.1. It can be concluded, that **3c** is slightly more toxic than tacrine for human hepatic stellate cells (HSCs).

#### 2.2.5. Neuroprotection Against Oxidative Stress

Neuroprotective activity of novel compound **3c** against oxidative stress was investigated in three experiments. Compound **3c** was tested at concentrations based on the IC_50_ results from AChE and BuChE inhibition assays. The first experiment examined whether the novel compound acted against exogenous free radicals induced by H_2_O_2_. The compound was tested at concentrations of 0.01 µM, 0.1 µM, 1 µM, and 10 µM. Trolox was used as a positive control. Cells that were incubated only with H_2_O_2_ had a viability of 84.27%. Cells treated with trolox and with 10 µM, 1 µM, 0.1 µM, or 0.01 µM **3c** had viabilities of 99.22%, 99.38%, 98.62%, and 96.41% (neuroprotection of 95.05%, 96.03%, 91.22%, and 77.16%) and 13.68%, 78.97%, 90.01%, and 91.53% (neuroprotection of 0%, 0%, 36.48%, and 46.17%), respectively (Table 2). Compound **3c** had no neuroprotection at concentrations of 10 µM and 1 µM and some neuroprotective activity at concentrations of 0.1 µM and 0.01 µM. However, the effect was much lower than that of trolox. The highest neuroprotective activity was obtained at the concentration of 0.01 µM, which is comparable with the IC_50_ of the AChE inhibition assay (0.025 µM). Values of *p* ≤ 0.05 were considered statistically significant for **3c**, not for trolox (one-way ANOVA).

An R/O (Rotenone/Oligomycin A) mixture was used to induce mitochondrial reactive oxygen species (ROS) by inhibiting mitochondrial electron transport chain complexes I and V [27]. The pre-incubation assay tested whether compound **3c** exerted neuroprotective properties through the activation of endogenous antioxidant pathways. In the co-incubation test, it was investigated whether the compound was a free-radical scavenger [28]. In the pre-incubation assay, SH-SY5Y cells were incubated with **3c** in a range of concentrations (0.0001–0.1 µM) for 24 h. The R/O mixture was then added to the cells and incubated. Trolox was used as a reference compound. Without the presence of **3c**, i.e., only the R/O mixture, cell viability was 46.77%. Cells incubated with **3c** had comparable viability to those incubated only with trolox at the same concentrations (Table 2). Of the cells treated with compound **3c**, only the 0.0001 µM concentration demonstrated poor neuroprotection (2.84%), while 0.001 µM trolox demonstrated 2.15% neuroprotection; hence, **3c** appears to demonstrate very small neuroprotective activity through the activation of endogenous antioxidants pathways at very low concentrations. However, no statistically significant difference was observed between **3c** and the R/O-treated control (*p* ≤ 0.05; one-way ANOVA).

In the co-incubation assay, cells were incubated with compound **3c** in the concentration range 0.1–0.0001 µM. Compound **3c** was incubated together with the R/O mixture for 24 h. Cells exposed to the R/O mixture alone had a viability of 47.91%. Cells incubated with compound **3c** at concentrations of 0.001 µM and 0.0001 µM demonstrated higher viability than those incubated with trolox at the same concentrations (Table 2). Compound **3c** showed some neuroprotection at concentrations of 0.1 µM (10.99%), 0.01 µM (12.90%), 0.001 µM (22.11%), and 0.0001 µM (29.10%). Trolox demonstrated very low neuroprotection, with values of 4.67% at 0.01 µM and 5.60% at 0.001 µM. These results were not statistically significant (*p* ≤ 0.05; one-way ANOVA) [29]. It can be concluded that compound **3c** had low neuroprotective activity as a free-radical scavenger.

#### 2.2.6. Determination of Global Histone H3 Phosphorylation (Ser 10)

Histone modifications, such as γH2AX, H3S10ph, H3K9ac, and H3K56ac, are markers of deoxyribonucleic acid (DNA) damage. Ser10 in the histone H3 is phosphorylated by mitosis and gene transcription. An increase in phosphorylation occurs within G2/M progression. Several kinases, such as Aurora-B, phosphorylate histone H3 at Ser10. The reduction of phosphorylation of H3 at Ser10 occurs during DNA damage. With DNA damage, Aurora-B loses its kinase activity, which results in the decrease of histone H3 phosphorylation on M-phase cells. Compound **3c** did not cause any decrease in phosphorylation of H3Ser10 (1 µM: 108.69% of phosphorylation in comparison to 100% control, 0.025 µM: 105.64%). The amount of phosphorylated H3Ser10 was 26.75 µg/mg for controls, 29.08 µg/mg for 1 µM **3c**, and 28.26 µg/mg for 0.025 µM **3c**. DNA damage was not observed at any concentration [30,31].

#### 2.2.7. Hyaluronidase (HYAL) Inhibition Test

Inflammation is a natural response to pathogens, xenobiotics, or mechanical injury, but without appropriate treatment it can lead to the development of chronic diseases. Several enzymes (for example hyaluronidase) are involved in promoting inflammatory pathway, generating inflammatory mediators, or enhancing tissue damage. Hyaluronidase (HYAL) degrades hyaluronan: an important part of the extracellular matrix. Therefore, enzymes control the size and concentration of depolymerized hyaluronan chains. Depolymerization can begin pathological processes, for example, increasing endothelial permeability or remodeling tissue during allergic reactions [32,33,34]. The non-steroidal anti-inflammatory drugs (NSAIDs) are commonly used to treat inflammation, but due to their side effects, their prolonged intake should be limited. Therefore, a novel drug with anti-inflammatory properties was synthesized and tested.

The inhibition of novel **3c** was tested by spectrophotometric assay according to Michel et al. [35]. The tested **3c** compound showed some inhibitory activity towards hyaluronidase (IC_50_ 468.77 ± 1.44 µM), but a positive control had better inhibitory activity (IC_50_ 56.41 ± 0.78 µM). It can be concluded that **3c** possesses some inhibitory activity towards hyaluronidase, which may indicate its anti-inflammatory properties.

#### 2.2.8. The Yeast Three-Hybrid Technology (Y3H)

Initial experiments indicated that the bait (AChE) was unable to start the autoactivation process. Moreover, positive and negative controls showed results consistent with Matchmaker Gold Y2H system, enabling the development of the following experiment. The Y2H analysis was conducted to evaluate putative protein–protein interactions between bait (AChE) and any of the preys (A4, BACE-1A, MAO B, and MAOPT) that could interfere with the Y3H analysis. The lack of blue colonies on double dropout medium lacking tryptophan and leucine and supplemented with X-Gal and Aureobasidin A (DDO/X/A) agar plates suggest that none of the analyzed protein pairs interact in terms of Y2H system measures.

Then, the Y3H test was completed with the same combinations of bait and preys. Blue colonies, suggesting putative interactions, were observed on DDO/X/A plates in the case of AChE–BACE–1A protein pairs in the presence of all three tested hybrid ligands. All other combinations of bait and preys (AChE–A4, AChE–MAO B, and AChE–MAPT) gave negative results for all three tested hybrid ligands. The more stringent QDO/X/A agar medium did not disrupt interactions mediated by hybrid ligands. However, the interactions diminished after transfer to QDO/X/A agar plates without hybrid ligands. It could be concluded that the hybrid ligand-mediated interaction is maintained only for te AChE and BACE-1A protein pair.

The activity of β-galactosidase (88.69 ± 2.57 units) observed for **3c** was relatively low to moderate as compared to the results of other Y3H experiments. The activities of β-galactosidase observed for the dexamethasone-trimethoprim hybrid ligand was significantly higher (about 500 units) [36]. However, in the other Y3H system using the methotrexate-SLF hybrid ligand, the β-galactosidase activity was lower (about 150 units) and more in line with the values observed in our experiment [37].

#### 2.2.9. LD_50_ and Toxicity Class Prediction

To verify the LD_50_ and toxicity class of the new compounds before in vivo assay, we used ProTox-II, a webserver for the prediction of small molecule toxicity [38]. The results are shown in Table 3. All of the eight new synthetized compounds were predicted to have 10–52.5-fold higher LD_50_ than tacrine. Two of the new derivatives had the same LD_50_ to bis-7-tacrine. Moreover, tacrine was found to be in a lower predicted toxicity class (2) than all the new compounds and the second reference.

#### 2.2.10. In Vivo Acute Oral Toxicity

The purpose of the study was to assign a new tetrahydroacridine derivative to the acute toxicity class and to obtain LD_50_ cut-off values. This method allows for the determination of exposure ranges and acute toxicity of the tested substance. Moreover, it can observe signs of toxicity, even delayed ones. Tetrahydroacridine derivatives are synthesized on the basis of tacrine, a drug that has been used for treating AD for a few years. Due to the chemical modification, the new derivatives should possess lower toxicity and better efficacy.

Compound **3c** did not cause any deaths at the dose of 300 mg/kg at the first and second stages. Therefore, a dose of 2000 mg/kg was administrated. At the first stage of dose 2000 mg/kg, all three mice died. Compound **3c** can be classified as Category 4 of the GHS (Globally Harmonized System), with LD_50_ cut-off values of 500 mg/kg (Figure 3). LD_50_ of tacrine after oral administration to mice was 39.8 mg/kg [39].

In the LD_50_ and toxicity class prediction, compound **2c** was predicted to have a 1000 mg/kg LD_50_ and a 4 toxicity class. In the in vivo assay on animals, compound **3c** turned out to have a lower LD_50_ cut-off (500 mg/kg) than in the prediction calculations. However, the toxicity class was the same in both assays. When it came to comparison of LD_50_, obtained results were similar, but not exactly the same. More trustworthy are the in vivo results on real animals, where metabolism, absorption, and all pharmacokinetics process are ongoing in the body. The new compound had to pass through the whole LADME (Liberation, Absorption, Distribution, Metabolism, Excretion and Elimination) in the mice’ bodies, which is much more crucial information to researchers than is a computer prediction. Moreover, in vivo studies are always required for clinical studies. As we see in Table 3, prediction accuracy of LD_50_ for **2c** was not 100%, but 69.26%. Therefore, the difference between prediction and in vivo results is understandable.

#### 2.2.11. pKa and logP Assay

The physicochemical properties of an active compound are needed to investigate the ability to permeate biological barriers. Our compounds are novel potential anti-AD drugs and hence need to pass through the blood–brain barrier (BBB) to obtain therapeutic effect. One of the most important properties in BBB permeation is log*P*. Log*P* assay requires the use of an uncharged molecule during analysis; a p*K*a assay was also performed to obtain the most reliable results. The procedure was performed as described previously [40] based on an initial method described by Musil et. al. [41] to calculate p*K*a values of the most active derivative of tested compounds. We estimated two different p*K*a values for each ionized form in the pH range 5.6 to 12.4 by using a regression equation (Figure 4). Our results are lower than values suggested by ChemAxon software and ACD/Percepta (Table 4).

The method was performed according to Chao Liang [42], with modifications, with the aim of performing fast, simple, and cheap determinations of log*P* of the most active derivatives of our compounds. The coefficient of determination for the calibration curve was above 0.96. The log*P* value for our test compound was 4.994. This value confirms the fulfilment of Lipinski’s rule of five, and allows further advanced BBB permeation assays to be performed. Both computers calculated values higher than experimental values. (Table 4)

#### 2.2.12. ADMET (Absorption, Distribution, Metabolism, Excretion, Toxicity) Analysis

ADMET prediction was performed for **2c**, which was the most active derivative among our tested compound series. We used experimental values of log*P* and p*K*a_1_ obtained from the test described above. ADMET prediction was made using ACD/Percepta software version 14.0.0 (Advanced Chemistry Development, Inc., Toronto, Canada). ADMET prediction confirmed sufficient brain penetration for central nervous system activity. The logPS value was equal −1.3. Compound **2c** can penetrate brain tissue: logBB was equal to -0.07, with 0.0081 for the fraction unbound in plasma and 0.01 for the fraction unbound in the brain. Despite the promising results, the ACD/Percepta model does not provided data on reliability of blood–brain barrier permeation testing, which is why the exact predicted values should be treated carefully. The probability of a positive Ames test was 0.60 (reliability index = 0.44). Compound **2c** fulfils the Lipinsky’s rule of five, because it has smaller molecular weight than 500, the correct number of hydrogen donors and acceptors, log*P* value is lower than 5, and TPSA (Topological Polar Surface Area) is lower than 140. ADMET prediction confirmed that **2c** has a good profile as a potential anti-AD drug.

#### 2.2.13. Molecular Modeling

In order to explain the binding mode of the obtained compounds in acetyl- and butyrylcholinesterase, docking studies were performed. In the case of AChE, the tetrahydroacridine moiety presented very consistent orientation for all compounds, creating π–π stacking and catio–π interactions with Trp84 and Phe330 (Figure 5; Figure 6 and Appendix A). Additionally, the protonated nitrogen atom created a hydrogen bond with the carbonyl group of His440. The arrangement of this fragment was almost identical as in the case of bis-(7)-tacrine in the 2CKM crystal structure. Significant differences between the compounds were observed for the dichlorobenzamide fragment. Compound **3a** with a short linker, containing two carbon atoms, occurred in bent conformation, in which the dichlorophenyl moiety created numerous hydrophobic interactions at the border of the anionic site (Phe330 and Phe331) and peripheral anionic site (Tyr334, Tyr121, Tyr70, and Trp279). Extension of the linker by one carbon atom (compound **3b**) led to the conformation becoming extended. In this conformation, the dichlorobenzene fragment created CH–π interactions with Trp279 and hydrophobic interactions with Tyr70 and Tyr121. A hydrogen bond between the amide nitrogen atom and hydroxyl group of Tyr121 was also observed (Figure 5A). Compound **3c**, with the highest activity towards AChE, presented a similar arrangement to **3b** but a longer linker, which allowed beneficial π–π stacking to be created with Trp279 and Tyr70, as well as CH–π interactions with Tyr121 (Figure 6A). For compounds with five and six carbon linkers (**3d** and **3e**), the top-rated configurations presented a less beneficial, bent conformation, in which the possibility of creating a hydrogen bond with Tyr121 and π–π interactions within the peripheral anionic site were not observed, unlike compound **3c**. This explains the inactivity of these compounds. Compounds with a long linker (**3f**, **3g**, and **3h**) exist in an extended conformation, creating π–π stacking with Trp279 and Tyr70, as well as CH–π interactions with Tyr121. Carbon linkers were located along the cavity of AChE, forming hydrophobic interactions mainly with Tyr334.

In the case of BuChE, the tetrahydroacridine moiety was located almost identically as in AChE, creating π–π stacking with Trp82 in the anionic site and hydrophobic interactions mainly with Trp430, Tyr440, and Met437 ( Figure 5; Figure 6 and Appendix A). A hydrogen bond was also observed between the protonated nitrogen atom and carbonyl group of His438. When docking to the AChE, significant differences occurred in the arrangement of the dichlorobenzamide fragment. Compound **3a** presented a bent conformation, in which the dichlorophenyl moiety was oriented towards the acyl pocket, creating CH–π interactions with Trp231 and hydrophobic interactions with Val288, Phe398, and Phe329. For compound **3b**, with the highest activity against BuChE, consistent poses in extended conformation were observed. The dichlorophenyl fragment formed hydrophobic interactions with Tyr332 in the peripheral anionic site (Figure 5B). Compound **3c** occurred in a bent conformation, in which, in addition to the interactions analogous to the ones for compound **3a**, halogen bonded with His438 was observed (Figure 6B). Compound **3d** also presented a bent conformation, but poses were less consistent, and scores were lower. Similarly, inconsistent poses were observed for compound **3e**. Such docking results explained the inactivity of these compounds. Derivatives with a long carbon linker (**3f**, **3g**, **3h**) occurred in extended conformation, in which the dichlorobenzene ring reached the surface of the BuChE, creating hydrophobic interactions with Tyr282, Ile356, and Pro285. Their carbon linkers interacted mainly with Tyr332.

## 3. Materials and Methods

### 3.1. Chemistry

Reactions were monitored by thin layer chromatography (TLC) using pre-coated silica gel aluminum plates. The flash chromatography (Puri Flash 430, Interchim, Montluçon France) purifying technique was performed using silica gel 60 (Merck, Darmstadt, Germany). Melting points (mp) were determined using the electrothermal apparatus with open capillaries and were uncorrected. The ^1^H NMR and spectra were recorded with a BrukerAvance III 600 MHz (Bruker, Billerica, Massachusetts, USA) spectrometer with tetramethylsilane (TMS) as an internal standard using DMSO-d6 or methanol-d4 as solvents. Mass spectra and high-resolution mass spectra (HRMS) analysis were performed with the use of an Agilent Accurate-Mass Q-TOF LC/MS G6520B system with a dual electrospray (DESI) source (Agilent Technologies, Santa Clara, California, USA). The detector in the analysis was tuned in a positive mode with the use of the Agilent ESI-L tuning mix in high resolution mode (4 GHz). Infrared (IR) spectra were recorded on the Mattson Infinity Series Fourier transform infrared (FT-IR) (Mattson Instruments, Inc., Fremont, California, USA) spectrophotometer, in ATR. All the chemical reagents used in the synthesis were obtained from commercial sources and used without purification. All chemical reagents were form Sigma-Aldrich, Poznań, Poland.

The scheme for the synthesis of compounds **2a**–**2h** and **3a**–**3h** are shown in Scheme 1. Intermediates **1a**–**1h** were prepared as described previously [26].

### 3.2. Biological Evaluation

#### 3.2.1. In Vitro Inhibition Studies on *Ee*AChE and *Eq*BuChE

Cholinesterase activity assays of the synthesized compounds were determined as previously reported using tacrine as a reference compound [43]. The substrate of the reaction was acetylthiocholine iodide. The activity was measured using 0.4 mg/mL 5,5′-Dithio-bis(2-nitrobenzoic) acid (DTNB, Ellman’s reagent) dissolved in phosphate buffer (pH 8.0). Next, 10 µL of AChE from electric eel (2 U/mL) and BuChE from equine serum (4 U/mL) were added. Solutions of tested compounds and the reference were prepared in nine different concentrations and used in 14 µL aliquots. The reaction was initiated by the addition of 40 µL substrate (1 mM or 2 mM for AChE or BuChE, respectively). The solutions were mixed and investigated in triplicate for 20 min at 30 ºC. A mixture without any test compounds was used as a control. Changes in absorbance at 412 nm were recorded in 96-well microplates using a spectrophotometric plate reader (Synergy H1, Biotek). Data are expressed as mean ± SD of at least three independent experiments.

#### 3.2.2. Kinetic Characterization of *Ee*AChE Inhibition

Compound **3e** was selected and kinetic studies were performed similar to the enzyme inhibition assay. To obtain estimates of the inhibition type, reciprocal plots of 1/V versus 1/[S] were constructed. The increase of the absorbance values was measured with different inhibitor concentrations (25, 75, 100 nM) and without inhibitor for the proposed substrate concentrations (ATI; 0.05–0.5 mM). All processes were assayed in triplicate.

#### 3.2.3. β-amyloid Assay

Amyloid β peptide (1–42) (Aβ_42_) was purchased from Sigma Aldrich and dissolved in DMSO to give a concentration of 12.58 µM. The compounds under study were dissolved in phosphate buffer (pH 8.0) to final concentrations of 5, 25, 50, and 100 µM. The assays were performed in a black, flat bottom 96-well plate. Firstly, 10 µL of the samples with 10 µL of Aβ_42_ dissolved in phosphate buffer were incubated for 24 h at room temperature. Samples of peptides without inhibitors were used as controls. After 5 min incubation with the dye (20 µL of ThT), fluorescence was measured at 446 nm (excitation) and 490 nm(emission). The percentage inhibition of the self-induced aggregation due to the presence of **3e** was calculated by the following expression: 100-(IF_i_/IF_0_ × 100), where IF_i_ and IF_0_ correspond to the fluorescence intensities in the presence and absence of the tested compound, respectively [44]. The reported values were obtained as the mean ± SD for triplicates in three different experiments.

#### 3.2.4. Cytotoxicity

Human hepatic stellate cells (HSCs, Sciencell) were cultured in the Stellate Cell Medium (Sciencell) with 1% Stellate Cell Growth Supplement, 2% fetal bovine serum (FBS), and 1% penicillin/streptomycin solution (Sciencell). Cells were kept in the incubator (37 °C, 5% CO_2_). At the beginning of the experiment, cells were seeded in the 96-well plates at a density of 5 × 10^3^ cells per well and incubated in the incubator for 24 h. After the incubation period, medium was removed and HSCs were exposed to 100 µL of the compound solutions over a range of concentrations (10–0.1 µM) or nothing but culture medium (blank control). Cells were further incubated for 24 h. At the end, medium was removed, cells were washed with PBS, and 50 µL of the MTT solution (0.75 mg/mL) was added to the cells. Plates were kept in the dark for 2 h at 37 °C. Finally, the MTT solution was removed and 100 µL of DMSO was added to each well. Plates with DMSO were kept at room temperature for 10 min. After this time, 5 µL of Sorensen Buffer was added to each well. Plates were swayed and the absorbance was measured at a wavelength of 570 nm by a microplate reader (Synergy H1, BioTek). The cell viability was expressed as a percentage of the control values (blank control) [45,46,47].

#### 3.2.5. Neuroprotection Against Oxidative Stress

Cell culture: the SH-SY5Y cells (human neuroblastoma, European Collection of Cell Culture) were chosen for the neuroprotection assay of novel compounds against oxidative stress. Cells were cultured in the Ham’s 12:EMEM (1:1) medium (Sigma Aldrich), supplemented with 15% FBS (Biowest), 100 units/mL of penicillin, 100 mg/mL of streptomycin (Biological Industries), 1% Non-Essential Amino Acids (Biological Industries), and 2 mM glutamine (Sigma Aldrich).

MTT assays: cells were seeded in the 96-well plates at a density of 5 × 10^3^ cells per well and cultured in the incubator (37 °C, 5% CO_2_) for 24 h. After incubation, medium was removed and cells were exposed to 100 µL of compound solution at different concentrations. Then, cells were incubated for 24 h; after this time, SH-SY5Y cells were washed with PBS and 50 µL of the MTT solution (0.75 mg/mL) was added to each well. The plates were incubated in the incubator (37 °C, 5% CO_2_) for two hours. Then, MTT solution was removed and 100 µL DMSO was added to the wells. Plates were kept at room temperature for 10 min and then 5 µL of Sorensen Buffer was added. The plates were swayed, and the absorbance was measured by a microplate reader (Synergy H1, BioTek) at the wavelength of 570 nm. The experiments were repeated three times and cell viability was expressed as a percentage of the control values.

Neuroprotection: the potential neuroprotective activities of 3c against oxidative stress were determined in three experiments. In the first experiment, hydrogen peroxide (H_2_O_2_) at the concentration of 100 µM was used to generate exogenous free radicals. SH-SY5Y cells were incubated with compound 3c in a range of concentrations (10–0.01 µM) for 24 h. Then, H_2_O_2_ was added, and cells were incubated with the compound and H_2_O_2_ for the next 24 h. In the second and third experiments, a mixture of rotenone (30 µM) and oligomycin A (10 µM) (R/O) was used to induce mitochondrial reactive oxygen species. In the second test, SH-SY5Y cells were incubated with compound 3c in a range of concentrations (1–0.001 µM) for 24 h, before the addition of R/O. After incubation, the R/O mixture was added and SH-SY5Y cells were incubated with 3c for next 24 h. In the third experiment, SH-SY5Y cells were incubated for 24 h with the addition of compound 3c and the R/O mixture. After 24 h, 3c and the R/O mixture were added at the same time and cells were further incubated for 24 h. Trolox was used as a positive control. Cell death was measured by the MTT assay. Each experiment was performed in triplicate. Data were shown as the percentage of the reduction of MTT in regard to the non-incubated cells [28,29,40].

#### 3.2.6. Comet Assay

Cell culture: Human astrocytes (HA, Sciencell, USA) were grown in the Astrocyte Cell Medium (Sciencell) supplemented with 2% fetal bovine serum (Sciencell), 1% Astrocyte Cell Growth Supplement (Sciencell), and 1% penicillin/streptomycin solution (Sciencell). Cells were cultured in the incubator at 37 °C and 5% CO_2_. To start the experiment, cells were seeded at a density of 2 × 10^5^ cells/well in the 6-well plates and were kept in the incubator (37 °C, 5% CO_2_) for 24 h. After the incubation period, medium was removed. Then, cells were exposed to compound **3c** (1 µM and 0.025 µM) and to tacrine (1 µM), or nothing but control medium.

#### 3.2.7. Determination of Global Histone H3 Phosphorylation (Ser 10)

Cell culture: Human astrocytes (HA, Sciencell, USA) were grown in the Astrocyte Cell Medium (Sciencell) supplemented with 2% fetal bovine serum (Sciencell), 1% Astrocyte Cell Growth Supplement (Sciencell), and 1% penicillin/streptomycin solution (Sciencell). Cells were cultured in the incubator at 37 °C and 5% CO_2_. To start the experiment, cells were seeded at the density of 1.5 × 10^6^ cells/well in the flasks and were kept in the incubator (37 °C, 5% CO_2_) for 24 h. After the incubation period, the medium was removed. Then, the cells were exposed to compound **3c** (1 µM and 0.025 µM) or nothing but control medium for 24 h. Then, cells were collected and histone extracts from them were prepared.

The levels of histone H3 phosphorylation (Ser10) were determined using a Global Histone H3 phosphorylation (Ser10) colorimetric assay kit (Epigentek) according to the manufacturer’s instructions. In the assay, the phosphorylated histone H3 at ser10 is captured in the wells of a 96-well plate coated with an anti-phospho histone H3 antibody. The captured phosphorylated histone H3 (Ser10) is detected with a labeled detection antibody followed by a color development reagent. The amount of phosphorylated histone H3 is proportional to the intensity of absorbance, which is measured by the microplate reader at a wavelength of 450 nm.

#### 3.2.8. In Vivo Acute Oral Toxicity

Compound **3c** was given orally (by gavage) in fixed doses, 5, 50, 300, or 2000 mg/kg body weight, at each stage of the study. Compound **3c** was administered at an initial dose of 300 mg/kg, as its toxicity was unknown. According to the graph from the Organization for Economic Cooperation and Development (OCED) 423 guideline, six test stages might be performed. Three animals (female mice Balb/c) take part in each stage. The number of stages depends on the deaths of mice, which is the final test parameter. Results allowed for the classification of compound **3c** into a Globally Harmonized Classification System for Chemical Substances and Mixtures (GHS) category. Moreover, the LD_50_ cut-off values were determined. The research was carried out on the basis of the agreement of the Local Ethical Committee for animal testing in Łódź No. 56/115 ŁB/2018. Compound **3c** was administrated in a first dose of 300 mg/kg of body weight.

The testing protocol is as follows: After the initial dose of 300 mg/kg body weight, if 2–3 mice die, the compound is given at a lower dose (50 mg/kg). The lowest dose (5 mg/kg) is administrated if 2–3 animals die after the 50 mg/kg dose. If there are no deaths or only one mouse dies after administration of the 300 mg/kg dose, the compound is given again at the same dose (300 mg/kg). If there are no deaths again, the highest dose (2000 mg/kg) is given. After each dose, the mice are observed for 14 days. After the observation period, the animals are sacrificed by intraperitoneal administration of a lethal dose of pentobarbital sodium and subjected to a post-mortem examination [48].

#### 3.2.9. In Vitro Inhibition Studies on HYAL

The hyaluronidase inhibition assay was set by turbidimetry according to USP XXII-NF XVII, modified by Piwowarski, to the 96-wells plates test [49,50]. Tested compound solutions were freshly prepared before the assay. The protocol started with the addition of 20 µL of the tested compound in monosodium phosphate buffer and 40 µL of hyaluronidase solution (22.5 U/mL, Sigma Aldrich) to each well. The plate was kept in the dark (10 min, 37 °C). In the next step, 40 µL of hyaluronic acid solution (0.03%, Sigma Aldrich) in monosodium phosphate buffer was added to the wells and the plate was incubated again (45 min, 37 °C). Finally, 300 µL of bovine serum albumin (0.1%, Serva) in sodium acetate buffer was added to the wells and the mixture was kept at room temperature for 10 min. The assay was carried out in triplicate. Changes in turbidity were measured by a microplate reader (BioTek, Winooski, VT, USA) at 600 nm. Heparin (WZF, Polfa, Warszawa, Poland) was the positive control [35]. The inhibitory activity of the tested compound was calculated from the Equation (1):(1)$$\%\;\mathrm{inhibition}=100x\left(1-\left(\frac{A_{HA}-A_{AN}}{A_{HA}-A_{HYAL}}\right)\right)$$
where AHA is the absorbance of solution without the enzyme (positive control), AHYAL is the absorbance of solution without the tested compound (negative control), and AAN is the absorbance of solution with the tested compound.

#### 3.2.10. pKa Assay

Potassium dihydrogen phosphate, potassium hydroxide, and methanol (POCH, Gliwice, Poland) were used to prepare the buffer solution. Stock buffer solution was prepared from 500 mL 0.02M potassium dihydrogen phosphate solution by adding 500 mL methanol. Working buffers were prepared by titrating stock buffer by 0.1M potassium hydroxide solution in a water/methanol mix (1:1). The pH value was set from 5.6 to 12.4 by 0.2 pH value per step. The measurement of pH was performed at 23 ˚C using a Mettler Toledo FiveEasy pH-meter with Lab pH electrode LE438 (Mettler Toledo). Our tested compound solution was 5 µM solution in a methanol/water (1:1) mix.

Spectrophotometric measurement was performed in a 96-well plate by using a Synergy H1 microplate reader (BioTek) with Gen5 software (BioTek). The full assay consisted of 35 UV spectra measurements, one for each work buffer solution. For the assay, 180 µL sequent work buffer and 20 µL tested compound solution was added to each of the 35 wells. For the blank, each of the 35 wells was administered with 200 µL sequent work buffer only. The measurement was performed at 23 ˚C. The spectra range was set from 280 to 380 nm by 1 nm steps. Obtained spectra were used to obtain experimental values of p*K*a as described previously [40]. All calculations were performed at ratios of 336/310 nm and 310/336 nm to avoid influence of apparatus noise, with the mean value as the final result. Experimental results were compared with computer calculated values. Computer calculations of p*K*a values were performed by chemicalize.com online software (ChemAxon 2018, Budapest, Hungary) and ACD/Percepta version 14.0.0 (Advance Chemistry Development, Inc., Toronto, Canada).

#### 3.2.11. LogP Assay

Methanol (POCH) was used as the organic modifier. Demineralized water was purified in our faculty. Triethylamine (TEA) (Sigma Aldrich) was used to achieve pH 11 and a concentration of 30 mM in methanol and water solution, because of the basic character of the test compound. Ten isocratic mobile phases were used in the calibration and assay. The first mobile phase was a 50:50 mix of 30 mM TEA methanol solution and 30 mM TEA water solution. Each subsequent mobile phase contained 5% more 30 mM TEA methanol solution and 5% less of 30 mM TEA water solution. The last mobile phase was composed of a 95:5 mix of 30 mM TEA methanol solution and 30 mM TEA water solution. The basic properties of the mobile phase did not affect the retention times of neutral calibration compounds.

The compounds used to prepare the calibration curve are listed in Table 5. Calibration compounds were selected in terms of similarity in structure to the test compounds. Stock solutions of calibration compounds and test compound were prepared in the mobile phase with the weakest eluting power of about 1 mg/mL. The injection concentration was about 100 µg/mL, and the injection volume was 7 µL.

A Waters 600 HPLC system with photodiode array detector (PDA) and chromatographic column Xbridge C18 50mm x 4.6mm i.d., 3.5 µm (Waters) was used. The detector was set at a respective optimum absorption wavelength for each compound. Data acquisition and processing were performed on a Waters Millennium software.

The procedure was performed according to Chao Liang et al. [42] with modifications. To prepare the calibration curve (Figure 1), each calibration compound was eluted by all mobile phases. All obtained retention times (tR) were placed in Equation (2):(2)$$\log k=log{\left(\frac{tR-t_{0}}{t_{0}}\right)}_{}$$
where value t_0_ was the time until death and Log*k* values were used to calculated log*k_w_* values of each calibration compound by the linear regression method and extrapolation.

The test compound was eluted by all mobile phases. Retention times were used in the same mathematical equations as in calibration to calculate the log*k_w_* value. The Log*P* value was read from calibration curve.

#### 3.2.12. Molecular Modeling

Three-dimensional structures of synthesized compounds were prepared using Corina on-line (Molecular Networks and Altamira). Atom types were checked, hydrogen atoms were added, and Gasteiger–Marsili charges were assigned with Sybyl 8.0 (Tripos). Acetylcholinesterase from 2CKM and butyrylcholinesterase from 1P0I crystal structures were selected for ligand docking. These proteins were prepared in the following way: All histidine residues were protonated at Nε, the hydrogen atoms were added, and ligands and water molecules were removed. The binding site was defined as all amino acid residues within 10 Å from bis-(7)-tacrine for AChE and 20 Å from the glycerol molecule present in the active center of BuChE. Docking was performed with GoldSuite 5.1 (CCDC). The standard settings of the genetic algorithm with population size 100, number of operations 100,000, and clustering tolerance of 1 Å were applied. After the docking process, 10 ligand poses, sorted by GoldScore (for AChE) and ChemScore (for BuChE) were obtained. PyMOL 0.99rc6 (DeLano Scientific LLC) was used to visualize the results of docking.

#### 3.2.13. The Yeast Three-Hybrid Technology (Y3H)

The putative induction of interactions between hybrid ligands and five selected proteins was evaluated by the Y3H method. Plasmids containing AChE, A4, BACE-1A, MAO B, and MAPT were prepared by Gene Universal Inc. (Newark, DE, USA). The company maintained cDNA synthesis and cloning of inserts in a proper frame, such as the NdeI/BamHI fragments into the pGBKT7 (AChE) or pGADT7 (A4, BACE1A, MAO B, MAPT) vectors. Moreover, the yeast codon optimization was realized to facilitate the high expression of human recombinant proteins in *Saccharomyces cerevisiae*. The cDNA encoding for human AChE (GenBank M55040.1) was modified by removing the fragments for a signal peptide (aa 1–51) and the domain responsible for protein tetramerization (aa 578–611). The prepared pGBKT7-hAChE plasmid contained the cDNA fragment encoding for aa 52–577. The pGADT7-hA4 construct was prepared from the human amyloid A4 protein fragment (UniProtKB-P05067-1), encoding for the 42 aa protein (aa 672–713). The pGADT7-hBACE1A plasmid was built from cDNA for the extracellular, N-terminal domain (aa 46–457) of BACE1A (GenBank: AF204943.1). Therefore, fragments encoding for signal peptide (aa 1–21), pro-peptide (aa 22–45), transmembrane (aa 458–478), and cytoplasmic (aa 479–501) domains were removed from the pGADT7-hBACE1A plasmid. The pGADT7-hMAPT plasmid was obtained from the cDNA encoding for the entire protein (aa 1–758) (UniProtKB-P10636-1). Finally, the pGADT7-hMAO B plasmid was produced from the cDNA encoding for cytoplasmic domain (aa 1–489) of human MAO B. All prepared plasmids were used to transform competent *S. cerevisiae* cells as described in the manual of the Matchmaker Gold yeast two-hybrid system (Takara/Clontech, USA). The pGBKT7-hAChE plasmid was used to transform the Y2HGold strain. The remaining four plasmids were applied to transform the Y187 strain. Transformed yeast strain Y2HGold (pGBKT7-hAChE) was tested for bait autoactivation. All negative and positive controls described in the manual of the Matchmaker Gold yeast two-hybrid system were performed.

The putative protein–protein interactions between bait (AChE) and the four preys (A4, BACE1A, MAO B, and MAPT) were analyzed by the small-scale mating procedure (5mL) based on the Matchmaker Gold yeast two-hybrid system manual recommendations. The prepared mixture was plated on DDO agar plates containing Aureobasidin A (200 ng/mL) and X-α-Gal (40 µg/mL) (DDO/X/A) to screen for putative protein–protein interactions. Then, the small-scale mating procedure in the volume of 10 mL was performed in the presence of hybrid ligands. The hybrid ligands were added to the mating solution at the concentration of 10 µM to initiate the putative hybrid ligand-mediated protein interactions. Suspension of mated cells was plated on DDO/X/A agar plates containing 10 µM hybrid ligand. Blue colony was transferred onto the QDO agar plates containing Aureobasidin A (200 ng/mL), X-α-Gal (40 µg/mL) (QDO/X/A), and 10 µM hybrid ligand to confirm the presence of interactions. The obtained blue colony was spread on QDO/X/A agar plates without the hybrid ligand.

The yeast β-galactosidase assay kit (ThermoFisher Scientific, USA) was applied to quantitatively analyze the strength of the hybrid-ligand induced interactions.

To evaluate the β-galactosidase activity in the yeast culture, the yeast β-galactosidase microplate assay protocol (stopped) was used according to the manufacturer’s instructions. The appropriate kit was supplied by Thermo Fischer Scientific (USA) [55]. The lack of protein–protein interactions between AChE and BACE-1 was excluded in control tests. The yeast cells used in the assay were grown for 96 h (30 ⁰C) on liquid QDO medium, containing 150 μg/mL aureobasidin A, 10 μM ligand, and 40 μg/mL X-α-Gal. The optical densities at 660 nm and absorbance at 420 nm were evaluated by (Synergy H1, Biotek). In the quantitative β-galactosidase assay, the hybrid ligand **3c** was used. Each assay was performed in triplicate.

## 4. Conclusions

New conjugates of tacrine with 3,5-dichlorobenzoic acid derivatives were synthesized and their structures established. The presented compounds constitute a class of tetrahydroacridine derivatives with a wide range of biological activities. New drug candidates were designed and synthesized to find a multi-target directed ligand. The assumption was to develop derivatives able to simultaneously affect many key mechanisms of AD. The biological screening results demonstrate that most of the synthetic tetrahydroacridine derivatives exhibited good *Ee*AChE and *Eq*BuChE inhibition, except **3d** and **3e**. Four of the new compounds demonstrate selectivity towards BuChE and two towards AChE. The spectrum of biological activity changed with the change of spacer in the structure. Kinetic study on the mechanism of *Ee*AChE inhibition and molecular modelling of the selected derivative **3c** were consistent with mixed inhibition. In addition, **3c** was a better BuChE inhibitor than donepezil. Compound **3c** showed other promising properties: it demonstrated good inhibition of self-induced A*β*1-42 aggregation and displayed neuroprotective activity against oxidative stress. As confirmed in three experiments, **3c** could be considered as a free-radical scavenger. In addition, it possessed an LD_50_ value 10–52.5-fold higher than tacrine. The study identified the toxicity class of the new compounds before in vivo assay. Based on in vivo acute oral toxicity, **3c** was classified as Category 4 GHS, with LD_50_ cut-off values of 500 mg/kg. Computational ADMET profiles predict that **2c** has a good profile as a potential anti-AD substance, and the improvement of water solubility due the hydrochloride form indicates that **3c** is a promising anti-Alzheimer drug candidate. Unfortunately, **3c** has poor anti-inflammatory properties. The results of the Y3H test suggest that the examined hybrid ligand may mediate the interactions only between AChE and BACE-1A. Previous experiments in our lab suggested the interaction between AChE and BACE-1A was mediated by another cyclopentaquinoline-derivative hybrid ligand [40]. Presented data confirm the interaction between AChE and BACE-1 is mediated by hybrid ligand **3c** in the in vivo conditions. The moderate value of this interaction (88.69 ± 2.57 units) putatively depends on the structure of the active cleft of AChE and BACE-1 and the affinity of hybrid ligand binding to both enzymes.

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
