# Peer review of "New Tetrahydroacridine Hybrids with Dichlorobenzoic Acid Moiety Demonstrating Multifunctional Potential for the Treatment of Alzheimer’s Disease"

_ijms, 2020, doi:10.3390/ijms21113765_

Round 1

Reviewer 1 Report

The submitted manuscript provides a comprehensive view of the chemical, physicochemical, biochemical, pharmacological and toxicological properties of a group of newly synthesized hybrid compounds based on connection of tetrahydroacridine and dichlorobenzoic acid. The experiments included chemical synthesis, evaluation of the inhibitory effect on acetylcholine esterase and butyrylcholine esterase in vitro, characterization of interactions with these enzymes, determination of cytotoxicity and neurotoxicity using cell models, evaluation of DNA damage and testing on anti-inflammatory effect in vitro. The study also presents model and experimental data on the pKa and lipophilicity of the tested compounds and the results of molecular modeling of the interaction with acetyl and butyrylcholine esterase. The manuscript further summarizes the results of in vivo toxicity modeling targeted at toxicological categorization and the results of a toxicological in vivo study following oral administration of a selected compound. The presented results showed interesting characteristics in some of the tested compounds including one compound with a potency for further preclinical development. The manuscript seems to be prepared carefully but the reviewer has several recommendations and suggestions to improve the text.

Minor revision:

  1. The introduction of the manuscript needs some revision, as it contains some redundant data and, on the other hand, some information should be added. For example, comments on diagnostics of AD are not relevant. Information on the gene APOE4 are not important in relation to the study. Therefore, the reviewer suggests to delete columns on the lines 64-71 and 53-55. Some broader information should be added to the sentence about MTDL strategy (l. 77-78). This concept is interesting and intensively investigated. Therefore, only one sentence on it seems to be unsufficient. Similarly, more information should be provided on significance of AchE and differences of this enzyme in comparison with BChE. In addition, there is also no information on A4, MAO B and MAOPT regarding their significance in AD, even they were included in the study using Y3F method. Also, there is no broader information on tacrin, which is used as the comparator in several tests performed in the evaluated study.
  2. The part 2.2.4. should be named Hepatotoxicity evaluation in vitro or similarly. In fact, the test does not evaluate hepatoprotection as it does not include evaluation of protecting effect of the tested compound against a hepatotoxin. Therefore, it should be considered as a cytotoxicity evaluation. What does the abbreviation THA for reference compound in the section 2.2.4. mean? It is necessary to explain it in this part of the text.
  3. L. 260: What is meant by the term “potential AD drug”? Does it mean potential antiAD drug?
  4. L. 402: The “SH-SY5Y” should be presented as “SH-SY5Y cell line” or “SH-SY5Y cells”.
  5. L. 420: There is no information on amount and concentration of H2O2 used in the experiments on protection against oxidative stress in Materials and methods.
  6. L. 613-614: The sentence “Computational ADMET…candidate.” is not clear. It claims that compound 2c has a good profile as a potential AD drug and compound 3c is a promising anti-Alzheimer drug candidate. This seems to be confusing. In fact, both agents are the same compounds, only difference is that 3c is the hydrochloride form. Therefore, the compounds cannot be considered as two different potential drugs. It must be corrected and explained more clearly.     

Author Response

  1. The introduction of the manuscript needs some revision, as it contains some redundant data and, on the other hand, some information should be added. For example, comments on diagnostics of AD are not relevant. Information on the gene APOE4 are not important in relation to the study. Therefore, the reviewer suggests to delete columns on the lines 64-71 and 53-55. Some broader information should be added to the sentence about MTDL strategy (l. 77-78). This concept is interesting and intensively investigated. Therefore, only one sentence on it seems to be unsufficient. Similarly, more information should be provided on significance of AchE and differences of this enzyme in comparison with BChE. In addition, there is also no information on A4, MAO B and MAOPT regarding their significance in AD, even they were included in the study using Y3F method. Also, there is no broader information on tacrin, which is used as the comparator in several tests performed in the evaluated study.

Thank you for suggestion.. We have followed this recommendation. We have added information concerning with MTDL strategy, significance of AchE and differences of this enzyme in comparison with BChE, A4, MAO B and MAOPT and tacrine.

  1. The part 2.2.4. should be named Hepatotoxicity evaluation in vitro or similarly. In fact, the test does not evaluate hepatoprotection as it does not include evaluation of protecting effect of the tested compound against a hepatotoxin. Therefore, it should be considered as a cytotoxicity evaluation. What does the abbreviation THA for reference compound in the section 2.2.4. mean? It is necessary to explain it in this part of the text.

It has been done. We have now clarified “THA” in the text.

  1. 260: What is meant by the term “potential AD drug”? Does it mean potential antiAD drug?

Thank you for the suggestion. It has been done.

  1. 402: The “SH-SY5Y” should be presented as “SH-SY5Y cell line” or “SH-SY5Y cells”.

Thank you for the suggestion. It has been done.

  1. 420: There is no information on amount and concentration of H2O2 used in the experiments on protection against oxidative stress in Materials and methods.

Thank you for the suggestion. It has been done.

  1. 613-614: The sentence “Computational ADMET…candidate.” is not clear. It claims that compound 2c has a good profile as a potential AD drug and compound 3c is a promising anti-Alzheimer drug candidate. This seems to be confusing. In fact, both agents are the same compounds, only difference is that 3c is the hydrochloride form. Therefore, the compounds cannot be considered as two different potential drugs. It must be corrected and explained more clearly.     

Thank you for suggestion. Compound 2c was considered as active substance, and because of not enough good water solubility of 2c the hydrochloride form (3c) was considered as potential anti-AD drug candidate. It is a fact that 2c and 3c differ in form. Hydrochloride form had a better water solubility, because of the protonation and charge, but it was still the same active structure. Thank you for finding this inaccuracy in the text, because it was indeed a confusing sentence and now is more clear. The correction in text has been done.

Reviewer 2 Report

The manuscript lacks important data to evaluate the perspective of designed and prepared compounds:

  1. The molecular design strategy should be depicted, but it is not.
  2. Scheme 2 should be moved to p. 3 and connected with chemistry section. Some bond lengths and angles are invalid.
  3. Source of AChE/BChE should be clearly given in the whole manuscript. The use of eel or equine enzymes does not make sense today, because the truly human enzymes are available. The use of eel or equine enzymes usually gives misleading results for human activity. And thus, maybe lead structure 3c is not the lead. Truly human enzymes have to be used and determined as well.
  4. Kinetic section is weird, because BChE is listed, but data are missing.
  5. Hepatoprotection study is missing the rationale for tacrine, which have to be metabolized to be toxic. This have to be considered.
  6. The neuroprotection with lead 3c is very poor.
  7. The LD50 is no longer used predict compound safety (it is tailored for toxins), but maximal tolerated dose or NOAEL test is used. The LD50 prediction could be very misleading. The experimental data for 3c were done, but statement “3c is less toxic than tacrine after oral administration” is not correct, because toxicity of tacrine is erased after multiple dose and by metabolism.
  8. 5 resolution have to be improved.
  9. The “K” from pKa should be written in Italics, same for “P” from logP. Why only 2 parameters were calculated/determined? The CNS bioavalability have to be considered in general (e.g. DOI: 10.1021/acschemneuro.6b00029), otherwise it could be misleading information. The experimental logP predict compound to be poorly soluble for possible in vivo use.
  10. Many assays were done for compound 3c (including in vivo toxicity), but 2c was evaluated in physchem, ADMET and molecular modelling? It does not make sense.
  11. In vitro assays (e.g. AChE, BChE, kinetics, …) are missing important details on e.g. concentration of individual components and are below standards on the field.
  12. The molecular modelling was done on human BChE, but in vitro data were provided with equine enzyme, why? Both should be human.
  13. The conclusion section is not supported by the data in the manuscript.
  14. In summary, too many assays were provided without appropriate strategy. No one can distinguish, what was the basic idea of molecular design and what is the on-target and the off-target. Targeting of too many pathways does not bring much benefit.
  15. There are many typos in the whole manuscript as well.

Author Response

Referee #2

Comments and Suggestions for Authors

The manuscript lacks important data to evaluate the perspective of designed and prepared compounds:

  1. The molecular design strategy should be depicted, but it is not.

Thank you for suggestion. We have now clarified design strategy in the text.

  1. Scheme 2 should be moved to p. 3 and connected with chemistry section. Some bond lengths and angles are invalid.

Thank you for the suggestion. It has been corrected. 

  1. Source of AChE/BChE should be clearly given in the whole manuscript. The use of eel or equine enzymes does not make sense today, because the truly human enzymes are available. The use of eel or equine enzymes usually gives misleading results for human activity. And thus, maybe lead structure 3c is not the lead. Truly human enzymes have to be used and determined as well.

Thank you for the suggestion. It has been corrected.

  1. Kinetic section is weird, because BChE is listed, but data are missing.

Thank you for the suggestion. It has been corrected.

  1. Hepatoprotection study is missing the rationale for tacrine, which have to be metabolized to be toxic. This have to be considered.

Thank you for the suggestion. Tacrine metabolism and metabolites were described.

  1. The neuroprotection with lead 3c is very poor.

Thank you for your observation. Yes, neuroprotection was below expectations, however it should be noted, that in the co-incubation study and incubation with H2O2 compound has shown some neuroprotective activity at lower concentrations.

  1. The LD50 is no longer used predict compound safety (it is tailored for toxins), but maximal tolerated dose or NOAEL test is used. The LD50 prediction could be very misleading. The experimental data for 3c were done, but statement “3c is less toxic than tacrine after oral administration” is not correct, because toxicity of tacrine is erased after multiple dose and by metabolism.

Thank you for this comment. Sentence “3c is less than tacrine after oral administration” was removed.

  1. 5 resolution have to be improved.

Thank you for suggestion. Figure 5 was generated by STATISTICA software and graphical improved, because STATISTICA has no available quality change options in generating figures. Figure 5 meets all requirements imposed by the journal including DPI value and resolution. Increase this values would cause to increase size of file, which is unacceptable to the journal. We will contact the editor to improve the quality of the figures.

  1. The “K” from pKa should be written in Italics, same for “P” from logP. Why only 2 parameters were calculated/determined? The CNS bioavalability have to be considered in general (e.g. DOI: 10.1021/acschemneuro.6b00029), otherwise it could be misleading information. The experimental logP predict compound to be poorly soluble for possible in vivo use.

Thank you for suggestion. Corrections have been applied in text. ACD/Percepta software provides parameters logBB, logPS, fraction unbound in plasma and fraction unbound in brain based on experimental pKa, experimental logP, and series of calculated parameters include HBD, TPSA, MW or ClogD, that were used by the Wager et. al. in their study. Moreover, ACD/Percepta performs a predictions based on algorithms and comparison to similar, well known structures, that are medicines and fragments of these structures. LogBB, logPS, fraction unbound in plasma and fraction unbound in brain are final results of all this procedures, what can be easily compare with another substances and easily used to inference about probable mechanism of CNS distribution. LogBB is a parameter that inform about substance distribution between plasma and brain tissue. LogBB = 1.5 and more indicates that substance is concentrated in brain tissue. LogBB = -1.5 and less indicates that substance distribution is restricted to plasma. LogBB = 0 indicates that substance is evenly distributed between this two mediums. LogPS is a parameter that informs about Blood Brain Barrier (BBB) permeability. LogPS = -1 indicates that substance is able to free diffusion. LogPS = -5 indicates that substance is almost completely not able to BBB permeation. LogPS = -3 indicate the average permeability. 2c is an alkaline substance, that has a poor water solubility in neutral form of structure. The change of protonation of molecule due hydrochloride form causes a single positive charge on nitrogen atom, increase the polarity of compound and water solubility improvement without an influence on mechanism of action. Therefore the hydrochloride form (compound 3c) was considered as effective in vivo substance. Experimental logP 4.994 is a high value, but in general not only logP influence on effectiveness in vivo. Promazine (logP = 4.810) or chlorpromazine (logP = 5.410) used as standards in logP assay are well known CNS medicines and they are used in medicine today. LogP is a parameter of structure in its neutral form, and logP assay required the neutral form to correct logP determination. More detailed information about logP and pKa assays was placed in answer 10.

  1. Many assays were done for compound 3c (including in vivo toxicity), but 2c was evaluated in physchem, ADMET and molecular modelling? It does not make sense.

Thank you for suggestion. Compound 3c is a hydrochloride form of 2c. Physicochemical assays required a free base form of compound. Usage of hydrochloride form would result in false values of pKa and logP, because of character of this studies. LogP and pKa assay required the neutral form of compound, because they are a physicochemical properties of substance without modulation of charge. This was a reason to use buffer solution in HPLC logP assay. Moreover, usage a hydrochloride form in HPLC logP test may results by erroneous results by the partial disintegration of the hydrochloride form at high pH. It is very probable, that chromatograms would show two peaks, one for each form of structure. Similar problems could occur in pKa assay during following steps of changing the pH values, however it would be more confusing, because pKa assay was based on UV spectra analysis. It is very probable, that solution containing hydrochloride form of structure with forced positive charge and a mixture of neutral and partly protonated molecules would result an inconclusive UV spectra that cannot be used to further analysis. Hydrochloride form did not affect the pharmacophore. Usage the structure of free base form – an active substance - is better for computer predictions and molecular modelling, because software cannot predict when the hydrochloride salt will go to free base in vivo and then goes to other form of protonation. Moreover, software not allow to input structures in hydrochloride form for the reason I wrote above. Therefore compound 2c was used in physicochemical assays, and 3c in other biological studies.

  1. In vitro assays (e.g. AChE, BChE, kinetics, …) are missing important details on e.g. concentration of individual components and are below standards on the field.

Thank you for the suggestion. It has been corrected.

  1. The molecular modelling was done on human BChE, but in vitro data were provided with equine enzyme, why? Both should be human.

Human and equine butyrylcholinesterases reveal high percentage of sequence identity (89.4%), including the sequence of the active site. The crystal structure of human BChE was obtained and published but there is no available 3D structure of equine BChE. In this case it is necessary to build homology model of equine BChE. As butyrylcholinesterases from both sources are very similar, we decided to use human enzyme for docking studies. Such procedure enabled us to have the first insight into the binding of compounds. 

  1. The conclusion section is not supported by the data in the manuscript.

We are grateful to this deep critical analysis of our manuscript. We have modified the conclusion section. 

  1. In summary, too many assays were provided without appropriate strategy. No one can distinguish, what was the basic idea of molecular design and what is the on-target and the off-target. Targeting of too many pathways does not bring much benefit.

Thank you for the important comment. This study aims at providing a lot of information about the mechanism of action for our new compounds.

  1. There are many typos in the whole manuscript as well.

Thank you for suggestion. It has been considered.

Reviewer 3 Report

Czarnecka et al present a work finding hits with possible anti-AD therapeutic activity. The study is well described and executed, obtaining interesting results. Moreover, the study combines in vitro, in silico and in vivo techniques that in the opinion of the reviewer is something remarkable. However, the reviewer has some concerns and suggestions.

1 - The observed IC50 and selectivity of 3c for Ache is similar to other known inhibitors? Or it is better? It would be interesting if the authors can introduce this comparative analysis in the text.

2 - It would be interesting to known if 3c prevents aggregation or just modulate it. If the 23 -32 % inhibitory activity, is translated in the unique presence of Aβ monomers or, except oligomers, can be other aggregates: dimers, trimers, pentamers, etc.

3 - The authors should indicate the LD50 ranges of each Toxicity Category, at least of those indicated in the text. In Figure 4 it is described for Category4 , but ther est of categories there is not an explanations. Moreover, it is very important that the authors indicate the average similarity and prediction acuracy of Pro-ToxII predictions. These values determine at which degree a prediction is trustable.

4 - The same should be done for ADMET predictions with Perceptra. The authors should indicate how much the prediction is trustable, i.e, if the analyzed compounds enter into the applicability domain of the model or not.

5 - Why docking results are only showed for compounds 2b and 2c and not for compounds of the series 3? Related to that it is not clear why some analysis are reported for compound 3c and others only for compounds from familiy "2".

Reading the document, the reviewer can understand that compounds 2 and 3 are the same but with HCL for the series 3, but it should be clearly explained in the text. Preferably from the begining, because untuil the chemistry section it is nor clearly understood.

6 - The binding modes of the compounds whose pictures are not shoed in the main text, as the authors make references to them, should be at least added to the Supporting Information.

7 - Docking calculations are rigid in the sense that the ligand is allowed to move but the target not. Induced fit events can be only see adding the dynamics, of the target, to the equation. There are several ways to do it. Using ensembles instead of a single conformation of the target, allowing the movement of certain residues of the target during the docking or postprocessing the docking poses with short MD simualtions. It would be nice if the authors can implement some of these options, at least for the most promising compounds. 

Author Response

Comments and Suggestions for Authors

1 - The observed IC50 and selectivity of 3c for Ache is similar to other known inhibitors? Or it is better? It would be interesting if the authors can introduce this comparative analysis in the text.

Thank you for the suggestion. It has been corrected.

2 - It would be interesting to known if 3c prevents aggregation or just modulate it. If the 23 -32 % inhibitory activity, is translated in the unique presence of Aβ monomers or, except oligomers, can be other aggregates: dimers, trimers, pentamers, etc.

The performed assay is based on binding of Thioflavin T to the beta sheets of an aggregated amyloid peptide resulting in an intense fluorescent product. In the presence of Aβ42 ligand, this reaction is abolished resulting in decrease or total loss of fluorescence. In the study, a new derivative inhibits beta amyloid aggregation but does not completely prevent.

3 - The authors should indicate the LD50 ranges of each Toxicity Category, at least of those indicated in the text. In Figure 4 it is described for Category4 , but ther est of categories there is not an explanations. Moreover, it is very important that the authors indicate the average similarity and prediction acuracy of Pro-ToxII predictions. These values determine at which degree a prediction is trustable.

Thank you for the suggestion. The rest of the categories was added and similarity with predictions was described.

4 - The same should be done for ADMET predictions with Perceptra. The authors should indicate how much the prediction is trustable, i.e, if the analyzed compounds enter into the applicability domain of the model or not.

Thank you for suggestion. ACD/Percepta performs a predictions based on specific algorithms and comparison to similar, well known structures, that are medicines and fragments of these structures. Results of this software predictions in combination with experimental results of biological physicochemical studies allow to infer about the probable mechanism of action.

5 - Why docking results are only showed for compounds 2b and 2c and not for compounds of the series 3? Related to that it is not clear why some analysis are reported for compound 3c and others only for compounds from familiy "2".

Reading the document, the reviewer can understand that compounds 2 and 3 are the same but with HCL for the series 3, but it should be clearly explained in the text. Preferably from the begining, because untuil the chemistry section it is nor clearly understood.

Compounds from series 2 are free bases while compounds from series 3 are the salts (hydrochlorides). All biological assays were performed at the specified conditions. At such conditions the equilibrium between free base and conjugated acid is established. Both forms of compounds (free bases and conjugated acids) present the same binding mode. However, protonated forms create hydrogen bond with His440 (AChE) or His438 (BuChE), thus binding more strongly. As figures present acid forms (series 3), we changed the numbering of compounds.

6 - The binding modes of the compounds whose pictures are not shoed in the main text, as the authors make references to them, should be at least added to the Supporting Information.

We added the figures with binding modes of selected compounds to the supplementary information.

7 - Docking calculations are rigid in the sense that the ligand is allowed to move but the target not. Induced fit events can be only see adding the dynamics, of the target, to the equation. There are several ways to do it. Using ensembles instead of a single conformation of the target, allowing the movement of certain residues of the target during the docking or postprocessing the docking poses with short MD simualtions. It would be nice if the authors can implement some of these options, at least for the most promising compounds. 

Indeed, molecular dynamics simulations allow for full investigation of the ligand-protein interactions. However, such simulations are time-consuming. As we wanted to have the first insight into the ligand binding we performed only docking studies.

Round 2

Reviewer 2 Report

Authors improved a manuscript a lot. However, its scientific value seems to be average and far away from current knowledge on the field. The typos and invalid abbreviations should be corrected prior to acceptance.

Author Response

Thank you for comments. We've corrected typos, incorrect abbreviations, and added missing abbreviations.

Reviewer 3 Report

The reviewer thanks the authors for the answers. Now the manuscript is much better bur there are still some concerns that should be addressed.

Question 3.  The authors are not including the average similarity and predicted accuracy in the table or write a sentence explaining which is the accuracy of the models. That's a very important information. It gives you an idea of how good the prediction is; if you can trust the predicted values or not. The average similarity and predicted accuracy values are part of the Protox-II  web server output, so they are easy to take -

Question 4 is not answered. The reviewer is not asking what the authors answered. The applicability domain of a computational model is the level of confidence on their predictions, such as the predicted accuracy of Protox II. If you don't have this information, I mean you don't know how much trustable the predicted values are, the authors should indicate that the applicability domain of the model is not known and the resulting values should be treated carefully, or something similar.

In the adding paragrah in page 9 the authors state that in vivo results are more trustablel. Thta's true, but this affirmation is not supported wit the previous sentences in the paragraph. Please revise it and re-write properly.

Author Response

Question 3.  The authors are not including the average similarity and predicted accuracy in the table or write a sentence explaining which is the accuracy of the models. That's a very important information. It gives you an idea of how good the prediction is; if you can trust the predicted values or not. The average similarity and predicted accuracy values are part of the Protox-II  web server output, so they are easy to take –

We re-calculated the LD50 and included the average similarity and predicted accuracy in the table.

Question 4 is not answered. The reviewer is not asking what the authors answered. The applicability domain of a computational model is the level of confidence on their predictions, such as the predicted accuracy of Protox II. If you don't have this information, I mean you don't know how much trustable the predicted values are, the authors should indicate that the applicability domain of the model is not known and the resulting values should be treated carefully, or something similar.

Thank you for comments. As we wrote, ACD/Percepta models is based on similarity to other structures, but BBB model does not provide unequivocal value of reliability of tests. This information and the suggestion about careful treatment of exact prediction BBB results have been added to manuscript. The Ames test prediction description has been enhanced with data on the reliability index, because this model was able to provide these data. Unfortunately our compounds are unique to the ACD/Percepta models databases, because 1,2,3,4-tetrahydroacridine derivatives are not very popular subject of research and at the same time an innovative approach to the search for new medicinal substances. Therefore most similar structure to compound 2c had similarity index 0.57, which explains the obtained reliability index values.

In the adding paragrah in page 9 the authors state that in vivo results are more trustablel. Thta's true, but this affirmation is not supported wit the previous sentences in the paragraph. Please revise it and re-write properly.

It is corrected. We have re-written this part of the manuscript.